# Comparison of Medical Comorbidity between Patients with Normal-Tension Glaucoma and Primary Open-Angle Glaucoma: A Population-Based Study in Taiwan

**DOI:** 10.3390/healthcare9111509

**Published:** 2021-11-05

**Authors:** Wei-Yang Lu, Ci-Wen Luo, Shyan-Tarng Chen, Yu-Hsiang Kuan, Shun-Fa Yang, Han-Yin Sun

**Affiliations:** 1Institute of Medicine, Chung Shan Medical University, Taichung 402, Taiwan; wesley1126@gmail.com (W.-Y.L.); kkjj88440@gmail.com (C.-W.L.); ysf@csmu.edu.tw (S.-F.Y.); 2Department of Ophthalmology, Changhua Christian Hospital, Changhua 500, Taiwan; 3Department of Pharmacology, School of Medicine, Chung Shan Medical University, Taichung 402, Taiwan; kuanyh@csmu.edu.tw; 4Department of Pharmacy, Chung Shan Medical University Hospital, Taichung 402, Taiwan; 5Department of Ophthalmology, Chung Shan Medical University Hospital, Taichung 402, Taiwan; shyan@csmu.edu.tw; 6Department of Optometry, Chung Shan Medical University, Taichung 402, Taiwan; 7Department of Medical Research, Chung Shan Medical University Hospital, Taichung 402, Taiwan

**Keywords:** normal tension glaucoma, primary open-angle glaucoma, comorbidities

## Abstract

The objective was to investigate different comorbidities developed in normal-tension glaucoma (NTG) and primary open-angle glaucoma (POAG) patients. This was a case-control study, with 1489 people in the NTG group and 5120 people in the POAG group. Patient data were obtained from the Longitudinal Health Insurance Database 2010 (LHID2010) of Taiwan for the 2008–2013 period. The chi-square test was used to compare categorical variables, such as gender, income and urbanisation level, between NTG and POAG patients, and the two-tailed *t* test was used to compare continuity between the two groups. We use a multivariate logic regression model to assess the risk of each participant. The results are expressed in terms of odds ratio (OR) and 95% confidence intervals (CI). Patients with NTG had significantly higher proportions of hypotension (adjusted OR, 1.984; 95% CI, 1.128–3.490), sleep disturbances (adjusted OR, 1.323; 95% CI, 1.146–1.528), peptic ulcers (adjusted OR, 1.383; 95% CI, 1.188–1.609) and allergic rhinitis (adjusted OR, 1.484; 95% CI, 1.290–1.707) than those with POAG. Conversely, arterial hypertension (adjusted OR, 0.767; 95% CI, 0.660–0.893), diabetes (adjusted OR, 0.850; 95% CI, 0.728–0.993) and atopic dermatitis (adjusted OR, 0.869; 95% CI, 0.763–0.990) had a lower risk in the NTG group than in the POAG group. We found that comorbidities such a hypotension, sleep disturbances and peptic ulcer and allergic rhinitis are more highly associated to NTG than POAG.

## 1. Introduction

Glaucoma is a disease that involves a progressive loss of retinal ganglion cells and causes irreversible blindness for people aged >60 years [1,2]. The elevation of intraocular pressure (IOP) that causes optic nerve damage and visual field loss is a major risk factor of glaucoma [3]. Primary open-angle glaucoma (POAG) is the most common type of glaucoma according to population-based prevalence studies worldwide; the estimated total prevalence for Europe was 2%, and the global prevalence was 2.2% [4]. In 1857, Von Graefe described a different kind of glaucoma that had typical glaucomatous optic neuropathy, but the intraocular pressure (IOP) was normal or lower than normal. Presently, this special form of glaucoma is called the normal-tension glaucoma (NTG). NTG is a multifactorial optic neuropathy which is characterized by progressive retinal ganglion cell death and glaucomatous visual field loss, similar to POAG. The major distinction of NTG from POAG is that the IOP does not exceed the normal range. Vascular dysfunction and ischemia have been considered as important factors in the progression of NTG [5]. In West Africa and Congo, the percentage of NTG among patients with POAG was only 6.3% [6] and 2.36% [7], respectively. In central Sweden, it was reported as 23.5% [8], which was comparable to its prevalence in Hong Kong (26.5%) [9]. The prevalence of NTG is higher in Asian populations than in Caucasian (30–38.9%) [10] and African American populations (57.1%) [9]. The aging population of China and systemic vascular disease could potentially explain the high prevalence of NTG in the Asian population [11]. However, systemic vascular diseases were more common in Asian populations, especially in Japanese and Chinese populations [12,13].

The elevation of IOP above 21 mmHg is an important indicator for retinal ganglion cell damage and visual field defect in glaucoma. Previous studies have shown that POAG is associated with cardiovascular, immunologic, metabolic, neurodegenerative and psychological diseases [14,15,16,17,18,19]. However, NTG patients with progressive retinal nerve fibre loss have a lower IOP, and abnormal ocular blood flow induces optic nerve dysfunction. Non-IOP-related cardiovascular dysregulation factors, such as systemic hypertension, systemic hypotension, nocturnal hypotension, and cardiac arrhythmia are implicated in NTG. The objective of our study was to investigate whether patients with these comorbidities are at higher risk of NTG or POAG using the National Health Insurance Research Database (NHIRD) in Taiwan.

## 2. Materials and Methods

### 2.1. Data Source

Taiwan’s NHIRD contains insurance data of ~98% of Taiwan’s population, which comes from the National Health Insurance (NHI) program of the National Institutes of Health. Since its implementation in 1995, the NHI program has provided comprehensive medical care to approximately 99% of the 23 million citizens and includes their outpatient records and inpatient treatment records. The data used in this study come from one of the NHIRD data subsets, the Longitudinal Health Insurance Database 2000 (LHID2000), which contains all claims data for 1 million representative beneficiaries randomly selected from the NHIRD from 2002 to 2013. These data can be linked to government surveys or other research data sets. Although only a small number of validation studies with small sample sizes have been conducted, they have widely reported a positive predictive value of >70% for various diagnoses. Currently, patients cannot opt out of the database, although this requirement is under review. NHIRD is a large and powerful data source for biomedical research [20]. Because this may be considered a major ethical issue when using databases for clinical research, the National Human Rights Agency also limits the amount of data required by researchers to less than 10% of Taiwan’s population. This means that researchers can only request data from approximately 2.3 million people at most. NHI also provides three longitudinal health insurance databases (LHID2000, LHID2005, and LHID2010), which randomly selected 1 million beneficiaries from the original NHIRD in 2000, 2005, and 2010, respectively [21]. The study protocol was reviewed and approved by the Institutional Review Board of Chung Shan Medical University Hospital. The database contains information on inpatient and outpatient visits, for which data are coded by clinicians according to the International Classification of Diseases, Ninth Revision, Clinical Modification (ICD-9-CM).

### 2.2. Study Population

In this nested case–control study, we excluded from the case group patients younger than 20-years-old. The literature has pointed out the incidence of NTG and POAG in young adults, but seldom pointed out the difference of comorbidities between NTG and POAG. Therefore, we believe that it is necessary to include participants over 20-years-old [22,23]. This study is a case-control study. The cases were a group of people who were diagnosed with NTG (ICD-9: 365.12) before 2013 and did not have POAG. The comparisons were a group of people with POAG (ICD-9: 365.11) who were not diagnosed with NTG before 2013. Patients with NTG or POAG had more than one outpatient diagnosis or one discharge diagnosis in the ophthalmology department. There were 1489 people in the case group and 5120 people in the comparison group (Figure 1).

### 2.3. Comorbidities

Our study used the following comorbidities, which were defined by more than three outpatient records or more than three hospitalisation records before the date of NTG or POAG diagnosis, as confounding factors for logistic regression adjustment: arterial hypertension (ICD-9: 401.x), hypotension (ICD-9: 458.9), sleep disturbances (ICD-9: 780.57), ischemic stroke (ICD-9: 434.11), Alzheimer’s disease (ICD-9: 331.0), diabetes (ICD-9: 250.x), Parkinson’s disease (ICD-9: 332.x), congestive heart failure (ICD-9: 428.x), peripheral vascular disease (ICD-9: 433.x), atrial fibrillation (ICD-9: 427.31), headaches(ICD-9: 784), migraines (ICD-9: 346), epilepsy and recurrent (ICD-9: 345), rheumatoid arthritis(ICD9: 714.0), systemic lupus erythematosus (ICD-9: 710.0), chronic kidney disease (ICD9: 585), hepatitis B (ICD-9: 070.2, 070.3, V02.61), tuberculosis (ICD-9: 010.x–017.x.), peptic ulcer (ICD-9: 533), depression (ICD-9: 311), malignant disease (ICD-9: 14x–23x), allergic rhinitis (ICD-9: 477.9), allergic conjunctivitis (ICD-9: 372.14), atopic dermatitis (ICD-9: 372.14) and fluid, electrolyte and acid–base disorders (ICD-9: 276.x).

### 2.4. Statistical Analysis

We used known confounding factors to enhance their comparability. The chi-square test was used to compare categorical variables, such as gender, income and urbanisation level, between NTG and POAG patients, and the two-tailed *t* test was used to compare the continuity between the two groups. We use a multivariate logic regression model to assess the risk of each participant. The results are expressed in terms of odds ratio (OR) and 95% confidence interval (CI). Using SAS 9.4 (SAS Analytical Solutions S.R.L., Bucharest, Romania) software data analysis, *p* value < 0.05 was considered statistically significant.

## 3. Results

Table 1 shows that there was no statistical difference between NTG and POAG patients in terms of gender and age. A Chi-square test was used to analyse the relationship between category variables. *p* < 0.05 was defined as achieving a statistical difference. We analysed the data with income and residency variables and found that 58.56% of NTG patients and 62.19% of NTG patients had low incomes; in terms of residence (rural vs urban), 43.38% of NTG patients and 36.84% of POAG lived in highly urbanized communities, showing a significant difference geographically. Among comorbidities, there were significant differences in the prevalence of arterial hypertension, sleep disturbances, diabetes, coronary heart disease (CHD), atrial fibrillation, headaches, migraines, peptic ulcers, malignant disease, allergic rhinitis and atopic dermatitis between them. The proportions of sleep disturbances, coronary heart disease, atrial fibrillation, headaches, migraines, peptic ulcers, malignant disease and allergic rhinitis were higher in NTG patients (29.62%, 19.01%, 1.95%, 30.76%, 4.1%, 24.71%, 8.8% and 28.95%, respectively) than in POAG patients.

Table 2 shows the risk of confounding variables for NTG development. The Wald test can be used to jointly test multiple hypotheses on multiple parameters. *p* value < 0.05 is considered to be of statistical significance. Multiple logistic regression was used to analyse POAG patients for comparison. The OR of low-income patients was 0.876 (95% CI: 0.773–0.992). In terms of the urbanisation level, compared with moderately urbanised individuals, the OR (odds ratio) of highly urbanised individuals was 1.399 (95% CI: 1.213–1.613). For comorbidities, the OR of arterial hypertension was 0.767 (95% CI: 0.660–0.893), the OR of hypotension was 1.984 (95% CI: 1.128–3.490), the OR of sleep disturbances was 1.323 (95% CI: 1.146–1.528), the OR of diabetes was 0.850 (95% CI: 0.728–0.993), the OR of peptic ulcers was 1.383 (95% CI: 1.188–1.609), the OR of allergic rhinitis was 1.484 (95% CI: 1.290–1.707), and the OR of atopic dermatitis was 0.869 (95% CI: 0.763–0.990).

Table 3 shows the risk of confounding variables for NTG in subgroups through multiple logic regression, and in comparison with POAG. The Wald test can be used to test multiple hypotheses on multiple parameters jointly in subgroup analysis. *p* value < 0.05 is considered to be of statistical significance. In female patients, the ORs were as follows: arterial hypertension, 0.727 (95% CI: 0.579–0.913); sleep disturbances, 1.296 (95% CI: 1.060–1.585); diabetes, 0.743 (95% CI: 0.589–0.938); peptic ulcer, 1.338 (95% CI: 1.076–1.665) and allergic rhinitis, 1.486 (95% CI: 1.216–1.816). In male patients, the ORs were as follows: sleep disturbances, 1.371 (95% CI: 1.113–1.688); peptic ulcer, 1.451 (95% CI: 1.172–1.796) and allergic rhinitis, 1.464 (95% CI: 1.201–1.785). In low-income patients, the ORs were as follows: arterial hypertension, 0.786 (95% CI: 0.647–0.954); sleep disturbances, 1.267 (95% CI: 1.053–1.525) and peptic ulcer, 1.420 (95% CI: 1.167–1.728). In non-low-income patients, the ORs were as follows: arterial hypertension, 0.726 (95% CI: 0.568–0.929); sleep disturbances, 1.451 (95% CI: 1.151–1.830); peptic ulcer, 1.341 (95% CI: 1.052–1.708) and atopic dermatitis, 0.740 (95% CI: 0.598–0.916). In patients living in highly urbanised areas, the ORs were as follows: hypotension, 3.870 (95% CI: 1.384–10.825); sleep disturbances, 1.390 (95% CI: 1.105–1.748); peptic ulcer, 1.306 (95% CI: 1.023–1.668) and allergic rhinitis, 1.434 (95% CI: 1.151–1.788). In patients living in moderately urbanised areas, the OR of allergic rhinitis was 1.550 (95% CI: 1.198–2.005). In patients living in an emerging town, the ORs were as follows: arterial hypertension, 0.588 (95% CI: 0.373–0.928); peptic ulcer, 1.722 (95% CI: 1.116–2.657) and allergic rhinitis, 1.839 (95% CI: 1.226–2.760). In patients living in a general town, the ORs were as follows: arterial hypertension, 0.369 (95% CI: 0.218–0.627); sleep disturbances, 1.642 (95% CI: 1.001–2.692) and peptic ulcer, 2.340 (95% CI: 1.394–3.927). In patients living in an aged township, the OR of arterial hypertension was 0.056 (95% CI: 0.005–0.634) and of sleep disturbances was 9.822 (95% CI: 1.395–69.157).

## 4. Discussion

According to the latest meta-analysis, the worldwide overall prevalence of POAG was 2.4%. However, there large variation exists about race. In our study, the prevalence of POAG is much lower than the previous study which may be explained by the fact that 37.5% of our population are less than 50 years old. The low prevalence of the elderly population may come from not being diagnosed thoroughly [22]. Our results showed that unlike in POAG, aging does not increase the risk of developing NTG. Although open-angle glaucoma mainly occurs after the age of 40, and adolescents have a certain risk [23,24]. There was no correlation between age and NTG through logic regression to adjust age, gender, etc., and subgroup analysis. This is a very interesting finding which represents the different pathogenesis of NTG and POAG. Compared with patients without NTG, a study by Chen et al., found that patients with NTG tended to be older and predominantly female [25]. These findings are not consistent with our study.

Our results indicated that the proportion of people with NTG in highly urbanised areas was higher than in moderately urbanised areas. Also, patients with higher income are shown to have a higher risk of NTG. The previous study demonstrated that an association between myopia and glaucoma was stronger in the NTG group. A study by Lee et al., proposed that the myopic NTG eyes have long-term IOP fluctuation which was significantly related to faster visual field progression [26]. Highly urbanised areas had a higher prevalence of myopia. Few studies have found oxidant stress to be associated with NTG [27,28]. Yilmaz et al., have shown that the total oxidant status and oxidative stress index values were significantly higher in the NTG group than in control groups [29]. A study indicated that areas with highly concentrated pollution are distributed among highly-developed cities in eastern China [30]. Previous studies have shown that oxidative stress and inflammation are potentially important mechanisms of intraocular pressure elevation caused by pollution and diet [31]. Noriko et al. found that decreased ocular blood flow is found to be related to oxidative stress, which contributes to the pathogenesis of NTG [32].

Our study indicated that hypotension, sleep disturbances, peptic ulcers and allergic rhinitis were significantly associated with a higher NTG risk. At the same time, arterial hypertension, diabetes and atopic dermatitis were seen less in the NTG group than in the POAG group. Previous studies showed that Alzheimer’s disease, vascular disease, obstructive sleep apnea (OSA)–hypopnoea syndrome were considered as the risk factors in NTG [33,34]. In the current study, a prevalence of hypotension is observed in NTG patients, in contrast to hypertension in POAG patients. Either high or low IOP could result in unstable vascular circulation in the eyeball, consequently affecting the stability of profusion pressure that may cause chronic oxidative stress [35,36] and induce mitochondria dysfunction of the optic nerve head [37]. A previous study indicated that insufficient blood supply of ocular blood flow, led to both low perfusion pressures and insufficient autoregulation, causing glaucomatous optic neuropathy [38]. As we know, low blood pressure is related to low ocular perfusion pressure, which is an important determinant of ocular blood flow. Low systemic blood pressure has been reported to occur more commonly in patients with NTG, causing progressive visual field loss [39,40].

Our data showed that NTG patients aged >50 years are at higher risk for sleep disturbances, regardless of gender. The patients with NTG living in an aged township showed higher odds ratios of sleep disturbances. Age could be a potential risk factor causing sleep disturbances in NTG patients. The association between glaucoma and OSA severity has been associated with structural and functional changes [41,42,43,44]. Recent studies have shown that OSA severity is highly prevalent in NTG patients [45]. Several studies found significantly decreased retinal nerve fiber layer (RNFL) thickness on optical coherence tomography in patients with moderate and severe OSA [46]. Higher OSA severity is associated with transient hypoxemia and obstruction of vasculopathy, which may may contribute to the pathogenesis of NTG [45,47].

Several researchers have reported the presence of an association between peptic ulcer and POAG [48,49]. There are some published data demonstrating the relationship between glaucoma and *Helicobacter pylori* infection in different countries, including Greece, Turkey, Iran, Korea, India and China [50]. Conversely, Galloway et al., indicated that *H. pylori* infection is not associated with glaucoma, including POAG, NTG and pseudoexfoliation glaucoma [51]. However, our data showed that NTG patients aged over 50 years old have greater risk of developing peptic ulcers than POAG patients, indicating that peptic ulcer is a comorbidity of NTG rather than POAG. Previous studies have shown that H. pylori infection is associated with arteriosclerosis-induced increased platelet activation and aggregation; thus, there may be a theoretical relationship between glaucoma and *H. pylori* [52]. Some glaucoma treatment medicines, such as carbonic anhydrase inhibitors like dorzolamide, may influence the pH level of the digestive system and thus potentially influence the infectivity of *H. pylori* [53].

The findings of this study showed a significant association between NTG incidence and allergic rhinitis regardless of age, gender and season. These results suggested that allergic rhinitis may have a role in the development of NTG. A previous report suggested that patients with allergic rhinitis and glaucoma have poor autonomic function [54]. It was found that patients with allergic rhinitis and glaucoma both had high levels of nitric oxide, which may accompany with increased endothelin-1 (ET-1) levels. Thus ET-1 and vascular dysfunction potentially play a key role in determining progressive visual field damage, this explains the relationship of allergic rhinitis and the normalised IOP of glaucoma [55,56,57,58,59].

This study has several potential limitations. First, laboratory data were lacking, so the patients’ biochemical markers, such as blood sugar and serum values, could not be collected. However, after adjusting for the confounding factors of comorbidities, this restriction did not affect our results. Second, NHIRD does not provide information on the visual acuity, OCT images or severity of visual field defects. Therefore, we could not evaluate if the severity rate of comorbidities was positively correlated with the severity of NTG. Third, the details of the patients’ usual lifestyle habits and family history, such as dietary habits, smoking and drinking, were not recorded in the health insurance database; therefore, how our results may have been influenced by these factors remains difficult to determine.

## 5. Conclusions

In conclusion, our population-based study is the first to compare comorbidities between large NTG and POAG cohorts using a large claimed database, and to offer potential explanations about a strong relationship between a medical illness and the risk of NTG. Based on this study, we could establish a conclusion regarding the role of hypotension, sleep disturbances, peptic ulcers and allergic rhinitis in NTG treatment. It is also hoped that in the future, when diagnosing the difference between NTG and POAG, judgments can be made on the basis of these comorbidities.

## Figures and Tables

**Figure 1 healthcare-09-01509-f001:**
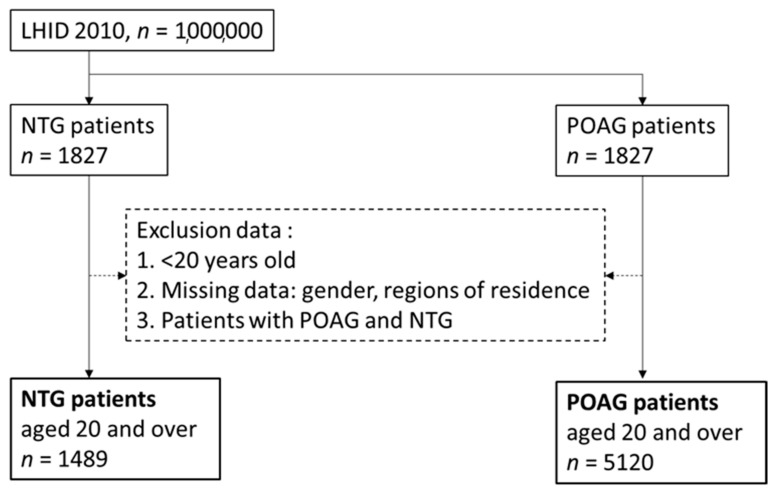
Study flowchart of NTG and POAG. NTG: normal-tension glaucoma; POAG: primary open-angle glaucoma.

**Table 1 healthcare-09-01509-t001:** Baseline characteristics of participants of POAG and NTG.

Confounding Variables	POAG (*n* = 5120)		NTG (*n* = 1489)		*p-*Value
**Gender**					
Female	2556	(49.92%)	713	(47.88%)	0.1664
Male	2564	(50.08%)	776	(52.12%)	
**Age**					
20–34 years old	794	(15.51%)	214	(14.37%)	0.2323
35–49 years old	1127	(22.01%)	312	(20.95%)	
50–64 years old	1519	(29.67%)	481	(32.3%)	
65 years old and over	1680	(32.81%)	482	(32.37%)	
**Low income**					
Yes	3184	(62.19%)	872	(58.56%)	0.0115
No	1936	(37.81%)	617	(41.44%)	
**Urbanization level**					
Highly urbanized	1886	(36.84%)	646	(43.38%)	0.0005
Moderate urbanization	1646	(32.15%)	408	(27.4%)	
Emerging town	712	(13.91%)	191	(12.83%)	
General town	537	(10.49%)	143	(9.6%)	
Aged Township	84	(1.64%)	20	(1.34%)	
Agricultural town	130	(2.54%)	42	(2.82%)	
Remote township	125	(2.44%)	39	(2.62%)	
**Comorbidities**					
Arterial hypertension	1671	(32.64%)	444	(29.82%)	0.0402
Hypotension	33	(0.64%)	22	(1.48%)	0.2590
Ischemic heart disease	87	(1.7%)	20	(1.34%)	0.3380
Sleep disturbances	1145	(22.36%)	441	(29.62%)	<0.0001
Ischemic stroke	46	(0.9%)	17	(1.14%)	0.3952
Alzheimer disease	4	(0.08%)	1	(0.07%)	0.8923
Diabetes	1200	(23.44%)	308	(20.69%)	0.0259
Parkinson’s disease	76	(1.48%)	17	(1.14%)	0.3231
Coronary heart disease	859	(16.78%)	283	(19.01%)	0.0453
Peripheral artery disease	67	(1.31%)	23	(1.54%)	0.4891
Atrial fibrillation	60	(1.17%)	29	(1.95%)	0.0223
Headaches	1375	(26.86%)	458	(30.76%)	0.0031
Migraines	134	(2.62%)	61	(4.1%)	0.0030
Epilepsy and recurrent	45	(0.88%)	13	(0.87%)	0.9830
Rheumatoid arthritis	87	(1.7%)	27	(1.81%)	0.7660
Systemic lupus erythematosus	2	(0.04%)	1	(0.07%)	0.6542
Chronic kidney disease	125	(2.44%)	40	(2.69%)	0.5939
Hepatitis B	159	(3.11%)	55	(3.69%)	0.2590
Fluid, electrolyte, acid–base disorders	34	(0.66%)	12	(0.81%)	0.5623
Tuberculosis	43	(0.84%)	15	(1.01%)	0.5418
Peptic ulcer	913	(17.83%)	368	(24.71%)	<0.0001
Depression	89	(1.74%)	23	(1.54%)	0.6104
Malignant disease	349	(6.82%)	131	(8.8%)	0.0095
Allergic rhinitis	1029	(20.1%)	431	(28.95%)	<0.0001
Allergic conjunctivitis	191	(3.73%)	47	(3.16%)	0.8363
Atopic dermatitis	1617	(31.58%)	448	(30.09%)	0.0045

Abbreviation: NTG: normal-tension glaucoma; POAG: primary open-angle glaucoma.

**Table 2 healthcare-09-01509-t002:** Logic regression analysis of NTG and POAG.

NTG		*p**-*Value
Confounding Variables	Adjusted OR (95%CI)	
**Gender (reference: female)**		
Male	1.111 (0.985–1.253)	0.0874
**Age (reference: 20–34 years old)**		
35–49 years old	0.957 (0.780–1.174)	0.6708
50–64 years old	1.137 (0.932–1.386)	0.2054
65 years old and over	1.038 (0.840–1.282)	0.7315
**Low-income (reference: No)**		
Yes	0.876 (0.773–0.992)	0.0368
**Urbanization level (reference: Moderate urbanization)**		
Highly urbanized	1.399 (1.213–1.613)	<0.0001
Emerging town	1.100 (0.904–1.337)	0.3417
General town	1.060 (0.851–1.319)	0.6047
Aged Township	0.992 (0.596–1.651)	0.9751
Agricultural town	1.257 (0.864–1.828)	0.2319
Remote township	0.950 (0.615–1.469)	0.8187
**Comorbidities (reference: without)**		
Arterial hypertension	0.767 (0.660–0.893)	0.0006
Hypotension	1.984 (1.128–3.490)	0.0174
Ischemic heart disease	0.656 (0.391–1.100)	0.1097
Sleep disturbances	1.323 (1.146–1.528)	0.0001
Ischemic stroke	1.276 (0.715–2.278)	0.4100
Alzheimer disease	0.727 (0.076–6.913)	0.7813
Diabetes	0.850 (0.728–0.993)	0.0400
Parkinson’s disease	0.674 (0.389–1.168)	0.1595
Coronary heart disease	1.139 (0.952–1.363)	0.1538
Peripheral artery disease	1.120 (0.683–1.835)	0.6542
Atrial fibrillation	1.511 (0.944–2.419)	0.0855
Headaches	1.023 (0.889–1.178)	0.7465
Migraines	1.296 (0.935–1.794)	0.1192
Epilepsy and recurrent	0.910 (0.483–1.713)	0.7702
Rheumatoid arthritis	0.895 (0.565–1.416)	0.6353
Systemic lupus erythematosus	1.646 (0.140–19.290)	0.6915
Chronic kidney disease	1.062 (0.726–1.554)	0.7574
Hepatitis B	1.057 (0.766–1.458)	0.7361
Fluid, electrolyte, acid–base disorders	1.086 (0.544–2.167)	0.8150
Tuberculosis	1.099 (0.600–2.011)	0.7598
Peptic ulcer	1.383 (1.188–1.609)	<0.0001
Depression	0.728 (0.450–1.178)	0.1964
Malignant disease	1.200 (0.964–1.494)	0.1022
Allergic rhinitis	1.484 (1.290–1.707)	<0.0001
Allergic conjunctivitis	0.778 (0.558–1.085)	0.1396
Atopic dermatitis	0.869 (0.763–0.990)	0.0350

Adjustment for gender, age, low-income, urbanization level, comorbidities. OR: odds ratio; CI: confidence interval. NTG: normal-tension glaucoma; POAG: primary open-angle glaucoma.

**Table 3 healthcare-09-01509-t003:** Logic regression analysis of NTG and POAG in subgroups.

Confounding Variables	Adjusted OR (95%CI)						
	Arterial Hypertension	Hypotension	Sleep Disturbances	Diabetes	Peptic Ulcer	Allergic Rhinitis	Atopic Dermatitis
**Gender**							
Female	0.727 (0.579–0.913) *	1.999 (0.898–4.452)	1.296 (1.060–1.585) *	0.743 (0.589–0.938) *	1.338 (1.076–1.665) *	1.486 (1.216–1.816) *	0.848 (0.707–1.017)
Male	0.816 (0.665–1.002)	1.786 (0.792–4.028)	1.371 (1.113–1.688) *	0.959 (0.776–1.184)	1.451 (1.172–1.796) *	1.464 (1.201–1.785) *	0.898 (0.744–1.085)
**Age**							
20–35 years old	0.375 (0.046–3.070) *	7.275 (0.679–77.911)	1.152 (0.630–2.109)	0.645 (0.216–1.927)	1.258 (0.587–2.697)	1.641 (1.144–2.352) *	0.994 (0.716–1.380)
35–49 years old	0.568 (0.361–0.894) *	2.128 (0.638–7.096)	1.208 (0.866–1.686)	0.795 (0.515–1.228)	1.252 (0.861–1.821)	1.596 (1.174–2.172) *	0.910 (0.685–1.208)
50–64 years old	0.775 (0.604–0.993) *	1.935 (0.634–5.906)	1.283 (0.994–1.657) *	0.914 (0.709–1.180)	1.508 (1.159–1.963) *	1.537 (1.184–1.995) *	0.711 (0.557–0.908) *
65 years old and over	0.870 (0.695–1.089)	1.565 (0.633–3.870)	1.522 (1.211–1.913) *	0.838 (0.664–1.057)	1.364 (1.083–1.718) *	1.342 (1.046–1.721) *	0.923 (0.730–1.165)
**Low-income**							
Yes	0.786 (0.647–0.954) *	1.718 (0.829–3.559)	1.267 (1.053–1.525) *	0.841 (0.692–1.022)	1.420 (1.167–1.728) *	1.435 (1.198–1.720)	0.956 (0.809–1.129)
No	0.726 (0.568–0.929) *	2.363 (0.930–6.003)	1.451 (1.151–1.830) *	0.871 (0.673–1.129)	1.341 (1.052–1.708) *	1.583 (1.263–1.985)	0.740 (0.598–0.916)*
**Urbanization level**							
Highly urbanized	0.816 (0.641–1.039)	3.87 (1.384–10.825) *	1.390 (1.105–1.748) *	0.911 (0.714–1.164)	1.306 (1.023–1.668) *	1.434 (1.151–1.788) *	0.831 (0.681–1.014)
Moderate urbanization	0.910 (0.689–1.203)	2.056 (0.692–6.104)	1.248 (0.956–1.629)	0.734 (0.544–0.991)	1.264 (0.949–1.684)	1.550 (1.198–2.005) *	0.974 (0.763–1.244)
Emerging town	0.588 (0.373–0.928) *	3.826 (0.996–14.699)	1.025 (0.663–1.584)	0.905 (0.574–1.427)	1.722 (1.116–2.657) *	1.839 (1.226–2.760) *	0.935 (0.640–1.365)
General town	0.369 (0.218–0.627) *	0.214 (0.022–2.118)	1.642 (1.001–2.692) *	1.094 (0.658–1.821)	2.340 (1.394–3.927) *	1.433 (0.875–2.347)	0.938 (0.600–1.466)
Aged township	0.056 (0.005–0.634) *	-	9.822 (1.395–69.157) *	0.131 (0.006–2.944)	1.402 (0.219–8.951)	11.113 (0.613–201.385)	0.309 (0.061–1.563)
Agricultural town	0.488 (0.176–1.353)	-	2.402 (0.845–6.828)	1.861 (0.632–5.481)	0.656 (0.198–2.179)	0.716 (0.236–2.175)	0.911 (0.292–2.842)
Remote township	1.145 (0.321–4.085)	-	0.851 (0.237–3.055)	0.933 (0.229–3.793)	0.456 (0.125–1.665)	0.714 (0.235–2.168)	0.404 (0.114–1.432)

Adjustment for gender, age, low-income, urbanization level, comorbidities. OR, odds ratio; CI, confidence interval. * *p* value < 0.05.

## Data Availability

No additional data are available.

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
