# Peer review of "Comparison of Medical Comorbidity between Patients with Normal-Tension Glaucoma and Primary Open-Angle Glaucoma: A Population-Based Study in Taiwan"

_healthcare, 2021, doi:10.3390/healthcare9111509_

Round 1

Reviewer 1 Report

The authors used a large, inclusive health database in Taiwan to explore co-morbid conditions for normal tension glaucoma.  They found a number of statistical relationship.  The validity of this finding is hampered, in my opinion, by three main issues:

  1. What is the basis for the “random” 1 million (~4% of the 23 million Taiwanese population)? Why not use all 23 million?
  2. The authors found an incidence of 5,120/1,000,000 for POAG. This incidence 0.5% seems lower than that typically reported in epidemiology studies. Is this a correct calculation?  If so, is this because a) secondary open-angle and closed angle glaucoma are not included, b) the incidence of glaucoma is lower in Taiwan than in the U.S. or other countries or c) the data or analysis is flawed.  The age of the POAG population – 37% of which are less than 50 years old makes this reviewer think that there might be some issue.
  3. Is OAG the correct “control” group of NTG? It seems that the general population of a similar age, or patients with other conditions such as migraine, hyperlipidemia, or gastrointestinal disease.

Author Response

Response to the Comments of Reviewer 1

Point 1: What is the basis for the “random” 1 million (~4% of the 23 million Taiwanese population)? Why not use all 23 million?

Response 1: Please find the interpretation shown in the session of “2.1 Data recourse” (Line 91-100).

Point 2: The authors found an incidence of 5,120/1,000,000 for POAG. This incidence 0.5% seems lower than that typically reported in epidemiology studies. Is this a correct calculation?  If so, is this because a) secondary open-angle and closed angle glaucoma are not included, b) the incidence of glaucoma is lower in Taiwan than in the U.S. or other countries or c) the data or analysis is flawed.  The age of the POAG population – 37% of which are less than 50 years old makes this reviewer think that there might be some issue.

Response 2: This is probably because that we used a higher criterion Patients with NTG or POAG had more than one outpatient diagnoses or one discharge diagnosis from the department of the ophthalmology. And which might Results in the lower incidence of POAG in this study compared to global prevalence.

Point 3: Is OAG the correct “control” group of NTG? It seems that the general population of a similar age, or patients with other conditions such as migraine, hyperlipidemia, or gastrointestinal disease.

Response 3: The major distinction of NTG from POAG is that the IOP does not exceed the normal range. Vascular dysfunction and ischemia have been considered as important factors in the progression of NTG. The key point that the reviewer pointed out is one of the questions that we would like to address in the paper as the different Medical Comorbidity may exist between the NTG and POAG groups.

Reviewer 2 Report

The study aims to assess the comorbidities present in POAG and NTG.

All glaucomatous patients older than 20 years were included. I do not understand this early cut-off, as open-angle glaucoma mainly occurs after the age of 40; furthermore, there is a real risk of including juvenile glaucoma, which also has malformative features that can confound the epidemiological data. 

Table 1 "Seasons". I do not understand the meaning of "seasons" in table 1: is it the period of onset or diagnosis of glaucoma? There is even statistical significance. Why? (See also Tab 2)

The higher correlation of NTG with the degree of urbanisation is also honestly incomprehensible to me.

Line 241 to 247.The considerations about corticosteroid glaucoma in treating atopic dermatitis seem disconnected from the context of the study and in my opinion should be removed from the paper. 

Author Response

Response to the Comments of Reviewer 2

Point 1: All glaucomatous patients older than 20 years were included. I do not understand this early cut-off, as open-angle glaucoma mainly occurs after the age of 40; furthermore, there is a real risk of including juvenile glaucoma, which also has malformative features that can confound the epidemiological data.

Response 1: Because this may be considered a major ethical issue when using databases for clinical research, our date recruit over 20 years old patients.

Point 2: Table 1 "Seasons". I do not understand the meaning of "seasons" in table 1: is it the period of onset or diagnosis of glaucoma? There is even statistical significance. Why? (See also Tab 2)

Response 2: One of the questions that we focused included the season of disease onset. However, this part has been removed from Table 1 & 2 to prevent confusing the audiences. Please see the changes made.

Point 3: The higher correlation of NTG with the degree of urbanisation is also honestly incomprehensible to me.

Response 3:In Taiwan, glaucoma is strongly related to the incidence of high myopia in clinics. Several studies have also demonstrated that there is a link between high myopia and NTG, and highly urbanisation is also one of the major reasons why high myopia is particularly significant in Taiwan. Please find the interpretations in Line 227 to Line 231.

Point 4: Line 241 to 247.The considerations about corticosteroid glaucoma in treating atopic dermatitis seem disconnected from the context of the study and in my opinion should be removed from the paper. 

Response 4:Thank you for the helpful comment, and this part has been removed from the manuscript.

Reviewer 3 Report

The authors investigate different comorbidities development in normal tension glaucoma and primary open-angle glaucoma patients, and they found that comorbidities such us hypotension, sleep disturbances and peptic ulcer and allergic rhinitis are highly associate to normal tension glaucoma than primary open-angle glaucoma.

  1. The final part of the abstract is incomplete
  2. L47 did you find some information about the reason of this distribution between different location in the world?
  3. The final part of the introduction seems that the no clear reason por normal tension glaucoma is defined. Some research orientates about these reasons, could you have mentioned some of them.
  4. In the results section on the tables, the p values do match with the first comparison? Or is the same p vale for all the comparison? Please clarify the p value along the tables.
  5. simplify table 3 legend and the composition of the table. It is impossible to read.
  6. Remove or resume the introduction information about general glaucoma on the discussion
  7. Focus the discussion only on the topic you find significative on the results do not speculate about other issues.
  8. Updated the reference within after 2005 at least, some of the were really old
  9. Reduced the number of references, more than 70 references for an original y excessive.

Author Response

Response to Reviewer 3 Comments

Point 1: The final part of the abstract is incomplete

Response 1: Please see the changes in the Abstract.

Point 2: L47 did you find some information about the reason of this distribution between different location in the world?

Response 2: yes, we did find some information in Line 57 to Line 60.

Point 3: The final part of the introduction seems that the no clear reason por normal tension glaucoma is defined. Some research orientates about these reasons, could you have mentioned some of them.

Response 3:Please see the revisions in Line 48 to Line 52.

Point 4: In the results section on the tables, the p values do match with the first comparison? Or is the same p vale for all the comparison? Please clarify the p value along the tables.

Response 4:Chi-square test is used to analyse the relationship between category variables. P<0.05 was defined as achieving a statistical difference. We also added in the result part.

Point 5: simplify table 3 legend and the composition of the table. It is impossible to read.

Response 5:Please find the change in the legend of Table 3.

Point 6: Remove or resume the introduction information about general glaucoma on the discussion. Focus the discussion only on the topic you find significative on the results do not speculate about other issues.

Response 6:Please see the revisions in the session of Discussion (Line202-217).

Point 7: Updated the reference within after 2005 at least, some of the were really old. Reduced the number of references, more than 70 references for an original y excessive.

Response 7:The References were updated. Please refer to the changes made in the manuscript.  

Round 2

Reviewer 1 Report

Thank you.  The authors have addressed my comments satisfactorily

Reviewer 3 Report

Comments solved

This manuscript is a resubmission of an earlier submission. The following is a list of the peer review reports and author responses from that submission.